# Mapping Lymph Node during Indocyanine Green Fluorescence-Imaging Guided Gastric Oncologic Surgery: Current Applications and Future Directions

**DOI:** 10.3390/cancers14205143

**Published:** 2022-10-20

**Authors:** Yiqun Liao, Jiahao Zhao, Yuji Chen, Bin Zhao, Yongkun Fang, Fei Wang, Chen Wei, Yichao Ma, Hao Ji, Daorong Wang, Dong Tang

**Affiliations:** 1Department of Clinical Medical College, The Yangzhou School of Clinical Medicine, Dalian Medical University, Dalian 116044, China; 2Department of Clinical Medical College, Yangzhou University, Yangzhou 225001, China; 3Department of General Surgery, Northern Jiangsu People’s Hospital Affiliated to Yangzhou University, Yangzhou 225001, China

**Keywords:** indocyanine green, sentinel lymph nodes, navigation, gastric cancer

## Abstract

**Simple Summary:**

The intraoperative navigation techniques develop rapidly, we present an update on current use and research on indocyanine green enhanced fluorescence imaging (ICG-FI) guided lymphangiography in gastric oncological surgery. ICG serves as a promising tattooed agent for lymphangiography, its application was restricted by the false negative lymph nodes and loss of specificity in targeting tumor cells. Besides, we summarized the current approaches for gain in sensitivity and specificity of ICG-FI guided lymphangiography in gastric oncological surgery. The next direction for reinforced ICG-FI was quantification of ICG intensity and detection of ICG distribution pattern, combining with intraoperative biopsy method.

**Abstract:**

Huge strides have been made in the navigation of gastric cancer surgery thanks to the improvement of intraoperative techniques. For now, the use of indocyanine green (ICG) enhanced fluorescence imaging has received promising results in detecting sentinel lymph nodes (SLNs) and tracing lymphatic drainages, which make it applicable for limited and precise lymphadenectomy. Nevertheless, issues of the lack of specificity and unpredictable false-negative lymph nodes were encountered in gastric oncologic surgery practice using ICG-enhanced fluorescence imaging (ICG-FI), which restrict its application. Here, we reviewed the current application of ICG-FI and assessed potential approaches to improving ICG-FI.

## 1. Introduction

Gastric Cancer (GC) is the most common gastrointestinal tract (GI tract) malignancy. Global cancer data for the year 2020 released by the International Agency for Research on Cancer (IARC) indicated that the morbidity and mortality of gastrointestinal malignancies were still high, with GC occupying fifth and fourth place, respectively [1]. Lymph node (LNs) metastasis drives the progression of gastric malignancy and indicates a poor prognosis [2]. The current intraoperative LN mapping techniques are aimed at providing a clear vision for evaluating regional LN drainage, guiding precise lymphadenectomy and thus gaining accurate tumour-staging information for gastric malignancies [3,4,5]. Accurate tumour staging will be helpful in the selection of adjuvant treatments and will facilitate decision-making processes with regard to radical gastrectomy [6,7]. In recent years, optical imaging technology has developed rapidly. The intraoperative navigation imaging technology represented by ICG-FI has made great advances in gastric oncologic practice. The fluorescence characteristic of ICG excited by near-infrared (NIR) light was used to tattoo LNs and achieve thorough and precise lymph node dissection [8]. Besides, the established concept of sentinel lymph nodes (SLNs) makes it applicable for surgeons to perform local resection of gastric cancer with the aid of ICG-FI. However, unpredictable false negative lymph nodes and the lack of specificity have limited the further application of ICG-FI. For now, many approaches have been carried out to overcome these obstacles, including multiple tracer methods integrated with ICG-FI as “Dual-tracer techniques”, organized ICG-FI with real-time biopsy methods, and enhanced imaging by artificial intelligence algorithms. This paper will review the current application status and future directions for ICG-FI in gastric cancer practice.

## 2. Current Applications of ICG-FI Guided Surgery

ICG-FI has recently been shown to be effective in locating SLNs in cancer patients and has exhibited a promising application prospect in the intraoperative navigation and localisation of LNs in a variety of cancers [9]. ICG is a non-toxic biological contrast agent that is excited by near-infrared light (750–810 nm) which produces enhanced NIR fluorescence, which can be detected, analysed, and imaged by the operation imaging device [10]. NIR light wavelengths (700 nm–900 nm) possess high tissue penetration (up to a few centimetres depth), and the autofluorescence of biological tissue excited by NIR is quite low, and thus provides sufficient tissue contrast [11]. In comparison to other biological dyes or visible fluorescence, ICG-FI has better tissue penetration, a more comprehensive dynamic range as well as minimal background fluorescence due to reduced scattering, ensuring that the judgement of the surgical visual field is freed from interference [12,13]. After being injected subcutaneously, ICG can quickly enter the small lymphatic vessels and tattoo lymphatic drainage with a fluorescence signal [14]. The rationale and in vivo images of ICG-FI are shown in Figure 1 [15]. The tattooed lymph system is finally presented as enhanced and clear lymphangiography.

### 2.1. Assessment of Sentinel Lymph Nodes to Achieve Tailored Partial Resection with Selective Lymphadenectomy

For cases of cT1 or cT2, the D2 or D1 scope of lymph node dissection seems to be unnecessary, not only because of excessive trauma and complications, but also because of compromised residual gastric function [16]. On the premise of ensuring the curative effect of early-stage gastric cancer, it is imperative to lower trauma, particularly for the preservation of residual gastric function. Miwa put forward a specific lymphatic basin dissection for gastric cancer [17]. Lymphatic basin dissection is defined as a SLNs biopsy method for the lymphatic system which is detected with dyed or fluorescent agent [18]. SLNs are defined as the first or more LNs that receive lymphatic drainage from the site of the primary tumour [19]. Cabanas first put forward the concept of SLNs and believed that the metastasis of tumour cells was carried out sequentially from the primary site of the tumour to the nearest site of lymphatic drainage [20]. Hence, the concept of SLNs is vital for the establishment of LN status in patients with early-stage cancer [21,22,23]. As ICG-FI was found effective in tattooing lymph systems, it emerged as a novel technique for SLNs detection in gastric cancer [24]. SLNs detection by utilizing ICG-FI received diagnostic accuracy 97.3% in T1 and 95.0% in T2 [25]. High detection rates (most > 90%) with good sensitivity were reported, particularly in cT1 GC patients [8,26,27,28,29,30,31,32]. Table 1 summarises the findings of SLNs detection studies in GC. When the SLNs biopsy was negative, function-preserving surgery with lymphatic basin dissection such as wedge (local) resection [33], segmental resection [34], minimal distal gastrectomy [18], or proximal gastrectomy [35] should be recommended. Schemas of the function-preserving curative gastrectomy with lymphatic basin dissection was shown in Figure 2, with respect to standard D2 or D1 gastrectomy, the content of nodal dissection in lymphatic basin dissection can be considered as D0 which means lower injure of surrounding tissue, controlled breakage of lymph nodes [18].

Although the prospect of limited surgery guided by ICG-FI was exciting, there are still worries in terms of oncological safety and standardization of SLNs biopsy procedure [43,44]. The SENORITA (SEntinel Node ORIented Tailored Approach) trial in Korea presented a short-term pathological and surgical outcomes investigation of laparoscopic sentinel node navigation surgery compared with laparoscopic standard gastrectomy with lymph node dissection and reported no significance of the rate and severity of complications in these two groups [45]. The long-term outcomes of limited lymphadenectomy in SLN-negative patients were investigated in a phase II trial, in which the safety and feasibility of laparoendoscopic-limited gastric resections without further lymph node dissections was confirmed with 3-year relapse-free and overall survival rates of 96% and 98%, respectively [46]. Another recent retrospective cohort study provided further evidence, indicating higher An overall survival (OS) rate of 96.8% and lower cumulative recurrence rate of 0.43% at 5 years in the sentinel node navigation surgery group than 91.3% and 1.30% of control group [18]. The limited surgery regarding SLNs biopsy resulted in advantages in long-term outcomes; however, further short-term and long-terms results are needed to determine the issue of oncological safety. On the other hand, methods of SLN biopsy are not uniform and vary in the use of tracers and imaging systems. We reviewed the injection and imaging protocol of investigated SLN biopsy studies, as listed in Appendix A. Theoretically, the appropriate injection of ICG is essential for technical standardization of SLN biopsies guided by ICG-FI. This means that, no matter what the route of injection (submucosal or subserosal) is, the tumor is infiltrated completely by the injected tracer [30,47,48]. It also has been elucidated that the dosage of ICG injection is the decisive factor of detection quality instead of the route of injection or duration between injection and surgery [49]. Further research needs to be done to demonstrate a technical standardization in terms of SLN to with regard to the route of injection (submucosal or subserosal) and dosage of tracers reported in the limited abovementioned studies.

### 2.2. ICG-FI Allows for Precise Lymphadenectomy in Gastric Oncologic Surgery

For resectable advanced GC, radical resection and adequate LNs retrieval are the cornerstone and pillar of GC oncologic surgery [50]. ICG-FI technique has been reported to increase lymph node harvesting in radical gastrectomies. A growing body of studies demonstrated increased LNs harvesting by ICG-FI techniques compared to conventional methods [51,52,53,54,55,56,57]. No statistical differences between the ICG and non-ICG groups in terms of perioperative, short-term, and long-term complications related to ICG injection and/or NIR imaging were reported [54,58]. The summary of those studies is shown in Table 2. Through effective lymphatic tattooing, ICG provided a clear vision of region lymph drainage and thus promoted the retrieval of extra LNs with no related complications and even lower blood loss [59]. It has been demonstrated that patients with ICG fluorescence had significantly higher numbers of LNs harvested in the subpyloric region [49,60]. Lymphadenectomy in the infrapyloric area was considered as a challenging procedure because the lymphatic and vascular drainage over the infrapyloric area was heterogeneous, and frequent LNs metastasis around this area was observed in advanced gastric cancer [61,62,63]. As a result, additional lymph node harvests in this area might improve the prognosis of patients. On the other hand, ICG-FI was considered as an effective tool in detection of LNs of splenic hilar with high negative predictive value, which is helpful in decisions of whether splenic hilar lymph nodes should be dissected [64]. In brief, ICG-FI is effective in revealing the drainage pattern of complex anatomical regions, which is helpful for operators to distinguish potential skip metastasis, so as to establish a confident and accurate lymph node dissection strategy and improve the prognosis of patients. The additional retrieval of LNs indicates ICG-FI as a powerful tool to facilitate complete lymphadenectomy and avoid damage to surrounding tissue, leading to improved prognosis. However, more high-quality large-sample RCTs are needed to validate this issue.

## 3. Limitations of ICG-FI in Current Gastric Cancer Practice

### 3.1. Absence of Specificity Contributes Additional Lymph Nodes Harvest but Loss in Precision Control of Gastric Cancer

It is generally believed that gastrectomy with more lymph nodes removal can reduce the recurrence and improve the survival of gastric cancer patients [67,68]. Some researchers have suggested that the ratio of metastatic lymph nodes is a sensitive independent prognostic factor, regardless of the number of lymph nodes examined [2,69,70], while negative lymph node count may serve as a supplementary strategy for the present tumor-node-metastasis (TNM) classification to further improve prognostic prediction efficiency [71]. The lymph node ratio (LNR) is an excellent prognostic tool to assess overall survival in populations with limited lymph node resection and minimal numbers of lymph nodes obtained at surgery and/or pathology [72]. Therefore, a limited and tailored strategy of precise lymphadenectomy might be accomplished depending on the identification of LNR. However, ICG is only the tattooed agent but not a specific target agent for metastatic LNs [73]. It might explain the greater number of lymph nodes harvested in ICG-FI guided lymphadenectomy, in which the whole removal of tattooed lymphatic drainage occurs regardless of metastatic or negative LNs. Despite ICG-FI providing additive retrieved LNs and precise organizational boundaries for the attenuated injury of adjacent tissue, the absence of tumor cell specificity confers few benefits on further precision control of gastric cancer.

### 3.2. Unpredictable False Negative Lymph Nodes Restrict Application of ICG-FI

In gastric oncologic surgery practice, the issue of unpredictable false negative restricts the application of ICG-FI. The SLNs biopsy procedure is composed of the tracing of target lymph nodes and subsequent en bloc dissected SLNs identification [18]. Relative low-detection failures were reported in SLNs biopsy procedures of early gastric cancer (cT1 and cT2) while higher false-negative rates were observed in lymph node biopsies of T3 and T4 gastric cancer [25,32,74]. The various false negativity rates might be explained by the different histological examination methods. A compromised histological examination may result in unacceptable false negative results [38]. In addition, the use of a single tracer and only one plane of pathological examination has some limitations, which is related to the high false negative rate of 46.4% [38]. Multiplanes detection and combined use of pathological examination methods would make false negativity of ICG-FI guided sentinel lymph node biopsy fall within a reasonable limit for intraoperative examination [30]. For ICG-FI guided GC lymphatic drainage tattooing, the staining rate of regional LNs by ICG were not 100%, that is, each station (D1 and/or D2) might have unstained LNs. When the metastatic regional drainage LNs were not stained by ICG, these lymph nodes were considered as false-negative lymph nodes [56]. The false-negative lymph nodes could be attributed to the injection of ICG not being close enough to the tumour and/or inadequate sites of injection. Furthermore, massive tumours and/or invaded tumour cells can block lymphatic vessels, resulting in changes in lymphatic drainage, making ICG unable to enter corresponding LNs [75]. On the other hand, the lymphatic gastric system is diverse and multidirectional, and skip metastasis frequently occurs, which might be explain the failures encountered in ICG-FI guided lymphography [76]. The mechanism of the false-negative lymph nodes is exhibited in Figure 3. In summary, the issue of unpredictable false negative lymph nodes serves as the main obstacle for radical lymphadenectomy.

## 4. Future Directions for ICG-FI Guided Surgery

As ICG-FI emerged as an exciting tool in gastric oncological surgery, multiple studies were done to seek strategies, including the double tracer method, intraoperative biopsy techniques, and reinforced imaging by algorithm implementation in order to achieve gains in sensitivity and specificity.

### 4.1. Applications of Multi-Tracers Integrated with ICG-FI

In order to improve the effectiveness of tracer guided gastric lymphangiography, the combine use of multiple tracers was considered as an alternative, namely the “dual-tracer-guided techniques”. The summary of reported combined use of tracers integrated with ICG-FI are listed in Table 3. Mostly, those “dual-tracer-guided techniques” t integrated with ICG-FI are traditional biological blue dyes or radiocolloid tracers. Multiple studies in the literature have demonstrated an enhanced tracing performance in the dual tracer method using blue dye or radiocolloid with ICG [77,78,79]. Such agents have been frequently reported as causing allergic reactions, and concerns over radioactive safety. Recently, a new generation of fluorescent agent 5-aminolevulinic acid (5-ALA) was introduced whose metabolites are biosynthesized into protoporphyrin IX (PpIX) in the cytoplasm that generates red fluorescence when it was excited by specific wavelength radiation. The abnormal transport activity of mitochondrial membranes and enzymes of cancer cells drive the accumulation of PpIX, generating significant fluorescence signals [80,81]. This change of fluorescence signals can be captured and converted into a recognizable fluorescent image, targeting potential micro-metastasis in lymph nodes. This might compensate for the absence of specificity in targeting metastatic lymph nodes and preserve confirmed negative lymph nodes when using ICG-FI [82]. Another promising fluorescence agent, namely sodium fluorescein (SF), generates fluorescence at the wavelengths (400–550 nm) of light excitation, which do not overlap with the wavelength (750–810 nm) of ICG-FI. i.e., These two fluorescent agents do not interfere with each other in imaging, and SF can be a remedial measure for a contaminated surgical field that is caused by spillage or incorrect injection of ICG-FI. Besides, the distribution patterns of the two dyes are different. ICG is more likely enriched in the lymphatic system, whereas SF has more affinity for surrounding tissues [82]. This is not only helpful to distinguish the tissue boundary during lymph node dissection, but also helpful to find lymph nodes that are not stained by ICG, especially for unstained metastatic LNs, namely the false negative LNs. This study only provides insights on the feasibility of the combined use of the two dyes, further evidence is needed to examine the combined use of SF with ICG in terms of sensitivity and accuracy. Moreover, several studies have managed to utilize magnetic tracer combined with ICG-FI. Co-localization of SLNs by both ICG-FI and a magnetic tracer was regarded as an effective supplementary means to allow for precise transcutaneous node identification, accurate intraoperative assessments of sentinel nodes, and to avoid higher echelon node involvement [83,84]. The magnetometer probe of this system provides audible and numerical feedback with respect to the nodal tracer uptake, which is capable of detecting the closure of ICG distribution, indicating a potential location of false negative LNs. Recently, a study showed that adjusting excitation light and quantifying ICG fluorescence intensity with VISION SENSE (R) may reduce the false negative rate of SLN and improve the sensitivity of ICG in detecting SLN [85]. In this report, the number of ICG-positive lymph nodes changed at the adjustment of the intensity of excitation light released by VISION SENSE^®^, which was capable of adjusting of the intensity of the excitation light and the quantification of ICG fluorescence intensity. When only a small number of ICG positive lymph nodes were available, more ICG positive lymph nodes could be observed by increasing the intensity of the excitation light. In other words, by adjusting the fluorescence intensity, those potential positive lymph nodes located in the depth of tissue or tattooed by inadequate fluorescent molecules caused by failure of the ICG injection and/or advanced tumor that could not be observed under the excitation light of general intensity might be exposed by the enhanced intensity of the excitation light. This undoubtedly suggests the next step for avoiding false negative lymph nodes. However, this study only presented us with a case of patients in cT1bN0M0 early gastric cancer without investigating the correlation between quantification of ICG fluorescence intensity and intensity of excitation light. Further studies should be done to evaluate the feasibility and effectiveness of this method in improving sensitivity in gastric cancer, especially for advanced cases.

### 4.2. Intraoperative Biopsy Technologies Improve Diagnostic Yield of ICG-FI

In order to apply the SLN concept in gastric cancer surgery to avoid unnecessary lymph node resection, a negative diagnosis of metastatic LN should be done before gastrectomy [30]. The summary of potential intraoperative biopsy technologies that might improve diagnostic yield of ICG-FI were shown in Table 4. The establishment of rapid and accurate intraoperative pathology is critical for determining SLN metastases [88]. Instant histopathological examinations prominent for SLNs biopsy procedure have been introduced, such as one-step nucleic acid (OSNA) [89] and reverse transcription-polymerase chain reaction (RT-PCR) [90] techniques. These techniques provide biopsy information for SLNs biopsy procedures but require en bloc fresh lymph node tissue which means only lymph nodes that fluorescently tattooed and dissected are available for biopsy. In a word, the false negative lymph nodes cannot be recognized by these biopsy techniques after ICG-FI detection. A confocal laser endomicroscopy (CLE) that recognizes tumor-specific fluorescent antibodies to achieve real-time in vivo immunohistochemistry (IHC) has been reported in a gastric lymph node metastasis model [91]. Instead of requiring fresh en bloc lymph node, the biopsy of the lymph nodes relied on scanning probe-based or needle-based CLE, providing real-time and in vivo ‘immunohistochemistry’ of tissues targeted by fluorescent antibodies. Through real-time intraoperative scanning, the confirmed negative lymph nodes might be preserved and thoroughly removed and confirmed as positive lymph nodes, making the lymphadenectomy more precise. However, this biopsy method is based on a needle or probe, which means that only a limited number of lymph nodes can be detected at one time, potentially leading to a prolonged process of lymphadenectomy. Furthermore, with the deepening application of optical imaging techniques in gastric cancer practice, a study using optical coherence tomography (OCT) to co-register with fluorescence molecular imaging (FMI) has been done [92]. In this system, OCT can generate high-resolution, non-invasive, and cross-sectional imaging of interested tissue microstructures in real-time, providing FMI quantification information. Besides, it can perform noninvasive real-time lymph node biopsy without specimens [93]. Interestingly, current OCT is able to scan the extensive tissues continuously [94]. OCT could conduct a large-scale inspection of the suspicious metastatic nodules independently, including the unstained positive LNs. Besides, the FMI quantification information might provide the distribution information of fluorescent agents (like ICG), detecting the potential filling defects of ICG in lymphatic drainage which indicate the occurrence of false negative LNs. Currently, OCT in combination with near-infrared fluorescence molecular imaging has been studied in the assessment of coronary status, for instance, plaque [95,96,97]. The prospect of co-registered OCT and ICG-FI is promising and could bring gastric oncologic surgery into new areas.

### 4.3. Artificial Intelligence, Augmented Reality and Machine Learning Algorithms for Enhancement of ICG-FI

A lot of effort has been made to improve the imaging quality and processing efficiency of ICG-FI. In the past few years, there has been a growing interest in developing artificial intelligence (AI), augmented reality (AR), and machine learning algorithms for the enhancement of ICG-FI. In a gastrointestinal malignancies surgery, artificial intelligence (AI) based real-time analysis microperfusion (AIRAM) outperformed traditional parameter-based methods in terms of accuracy [100]. The well-trained AI in this study was able to classify microcirculation states, adapting for real-time processing. This is also applicable to ICG-FI guided lymph node mapping. A machine learning algorithm-based neural network might help in suppressing light scattering and maximising imaging penetration depth for in vivo near-infrared imaging [101]. The application of machine learning and artificial intelligence technology may realize the automatic evaluation of collected image data in the future. Both of these methods are helpful to accurately display tissue features that are initially indistinguishable to human eyes [102]. An AR-based fluorescence imaging system was developed to provide real-field fluorescence images by matching the fluorescence pathways with the visible light. The FOV discrepancy and inconvenience of a fluorescence image displayed on a screen and replaced by a direct view of the surgical field may be addressed by this system [103]. The real-field image provides a clearer view of the edge of the fluorescent marking range, allowing for a more precise dissection of lymph nodes.

## 5. Discussion

ICG-FI has been extensively studied in the last few years and has proven its safety and feasibility in gastric oncologic practice not only for guiding limited surgery in early-stage gastric cancer but also for precise lymphadenectomy in radical gastrectomy. For early-stage gastric cancer (cT1/cT2), the procedure of SLNs biopsy has been widely examined. Instead of directly performing standard D2 lymphadenectomy, a function-preserving curative gastrectomy with lymphatic basin dissection was proposed after SLNs were proven negative. This limited surgery regarding SLNs biopsy has received positive feedback in long-term outcomes. Regardless of insufficient evidence on short-term oncological safety, the limited gastrectomy reduced trauma and preserved partial gastric function, which might improve the quality of life and hospital stays. In advanced gastric practice, ICG-FI increased the number of lymph nodes retrieval, providing guidance for anatomically difficult regions, which aided surgeon to found potential skip metastasis as to establish robust and precise lymph node dissection strategy. Limitations were encountered in ICG-FI guided gastric oncology surgery, the issue of unpredictable false negative lymph nodes causes failure of thorough lymph node dissection and threatens patients’ lives. Besides, the lack of specificity drives the harvesting of additional lymph nodes but a loss in the further precision control of gastric cancer. Multiple studies aim to achieve gains in the sensitivity and specificity of the ICG-FI to deal with the complex gastric lymphatic system. In the last few years, the implementation of dual-tracer methods have exhibited exciting performance improvements in promoting advances in detection sensitivity. Among these agents, SF showed completely different fluorescence excitation wavelengths and tissue distribution patterns in contrasted to ICG-FI, which was helpful in the identification of the distribution pattern of ICG in lymphatic vessels, so as to detect possible ICG infiltration defects in lymphatic vessels and find potential false negative lymph nodes. Moreover, magnetic tracers had a similar advantage in distinguishing the distribution of ICG-FI. The magnetometer probe provides information about nodal tracer uptake, determining possible abnormal lymphatic drainage in which false negative lymph nodes often occur. On the other hand, auxiliary intraoperative biopsy technologies, such as OSNA and RT-PCR, may detect tumor micro-metastasis rapidly. However, these methods are limited with regards to the requirements of LN tissue, which cannot be used to evaluate the status of lymph nodes before dissection. Furthermore, the probe-based or needle-based CLE, provides real-time and in vivo tissue specific biopsy, but has a limited detection range based on the needle or probe, which may potentially prolong the progress of surgery. Last but not least, OCT could provide tissue-specific biopsies and detect the intensity of fluorescent molecules. The next generation of OCT, namely optical coherence tomography angiography (OCTA), uses motion contrast which provides visualization of vasculature at real-time acquisition rates [104]. This is also available for the practice of revealing abnormal lymph flow. Essentially, all these efforts are based on improving the overall imaging quality of ICG-FI or as a supplement, whether it is an auxiliary algorithm, a combination of multiple tracing methods, or intraoperative biopsy technology. As reported in the previous case study, the intensity of fluorescence directly determines the imaging details, potentially leading to an increased probability of finding false negative lymph nodes. However, the relationship between fluorescence intensity and excitation light intensity has only been qualitatively described without quantitative research. Besides, the dosage of ICG injection was recognized as the determinant in detection of SLN, indicating that ICG concentration is a factor associated with imaging quality. The co-registered OCT-FMI system observed a linear relationship between fluorescence molecular imaging intensity and dye concentration, but the correlation of fluorescence intensity with ICG concentration was still unclear. Therefore, the quantification of fluorescence intensity regarding the excitation light intensity and ICG concentration is the key for further optimization of imaging quality. On the other hand, solely improving imaging quality and expanding imaging details will not solve the loss in tumor specificity of ICG. Correspondingly, more potential lymph nodes will be detected, which brings challenges to decision-making in lymph node dissection. The intraoperative real-time biopsy technology can be used as a supplement to quickly determine the status of lymph nodes and formulate an accurate radical lymphadenectomy strategy. Moreover, skip metastases have been associated with tumor size and the presence of lymphatic invasion and reported with poor survival [105]. With additional potential lymph nodes detected by improved imaging efficiency of ICG, aberrant lymphatic drainage with changed ICG distribution pattern will more likely be implied. In this case, clarification of ICG distribution pattern is helpful to understand the internal mechanism of abnormal lymphatic drainage and further predict skip metastases in gastric cancer.

## 6. Conclusions

ICG-FI is effective in guiding gastric oncologic surgery, providing individualized surgical strategies for precision control of gastric cancer. The current central issue is that of overcoming the obstacle of unpredictable false negative lymph nodes. In this review, we put forward that the detection of the distribution patterns of ICG under the quantification of fluorescence intensity in gastric cancer lymphatic system may potentially address the issue of unpredictable false negative lymph nodes. Techniques like OCT, which not only provide real-time, wide-field, in vivo and non-invasive rapid biopsy, but also offer the quantification information of fluorescence agents might be an appropriate consideration for the reinforcement of ICG-FI. In order to achieve the next breakthrough for ICG-FI, further studies should be performed.

## Figures and Tables

**Figure 1 cancers-14-05143-f001:**
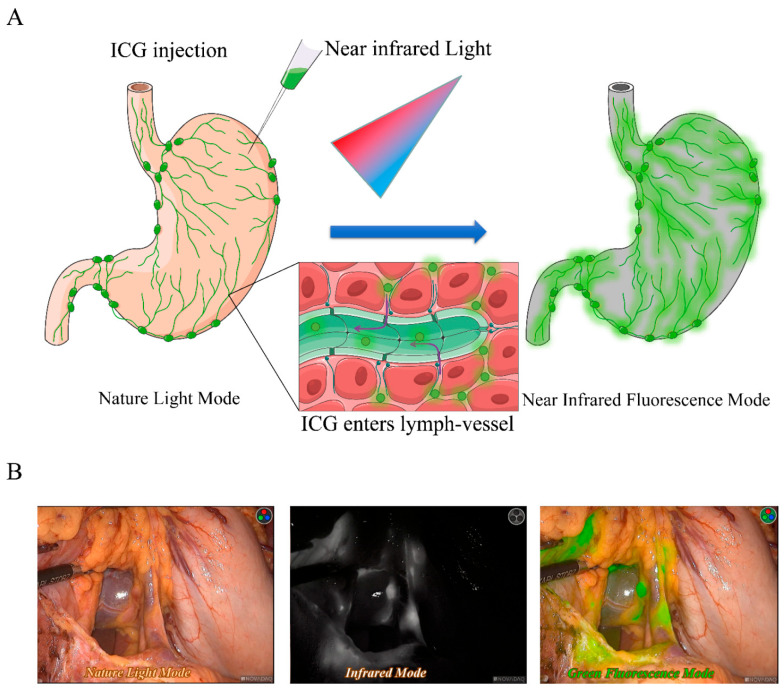
Illustration of ICG-FI: (**A**) rationale of ICG-FI. After injection, ICG enters and tattoos the lymphatic system, generating fluorescence under the excitation of near-infrared light. The Fluorescent signal can be captured and analyzed by image system which provides real-time in vivo images of lymphangiography; and (**B**) in vivo images of lymphangiography in gastrectomy. The picture on the left side represents the surgical field under the natural light. The picture in the middle presents the infrared mode of surgical field in which fluorescence is obvious compared to the almost invisible surrounding tissue. The picture on the right side shows the surgical field with both visible natural light and invisible near-infrared light. This mode allows for distinguishing tattooed lymph nodes and adjacent tissue. In vivo pictures are from supplementary video of Ref. [15].

**Figure 2 cancers-14-05143-f002:**
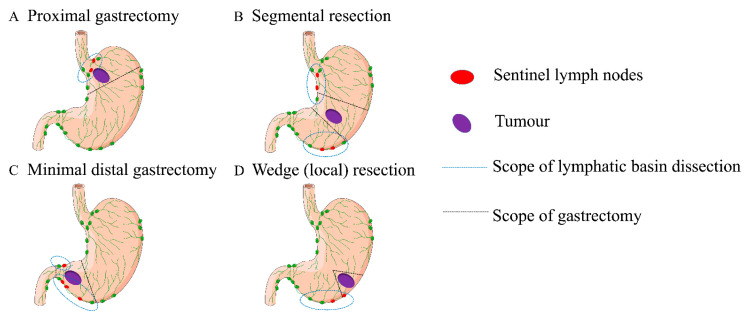
Schematic of function-preserving curative gastrectomy with lymphatic basin dissection. The red circle represents sentinel lymph nodes, the purple ellipse indicates tumour, the blue dotted line indicates scope of lymphatic basin dissection and the black dotted line indicates Scope of gastrectomy.

**Figure 3 cancers-14-05143-f003:**
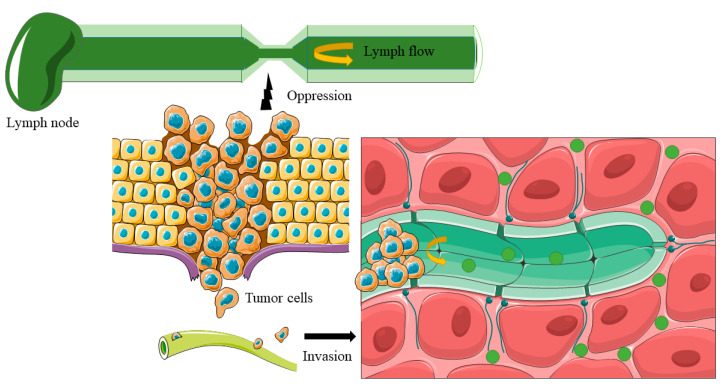
The mechanism of false-negative results as consequence of lymphatic obstruction. The false-negative results occurred when the tumor was oversized (>4 cm) that compressed the lymphatic vessels. The lymphatic reflux was stopped as well as ICG, causing the imaging failure of distal lymph nodes. On the other hand, the lymphatic reflux can be stopped when the tumor cells invaded the lymph vessels, which can also cause lymphatic obstruction.

**Table 1 cancers-14-05143-t001:** Summary of the studies on SLNs detection in GC. Ref: reference; FNR: false negative rate; LAG: Laparoscopic gastrectomy; OG: Open gastrectomy.

Ref.	NO. of Patients	Detection Rate	Sensitivity	Specificity	Considerations
Hiratsuka, M. 2001 [26]	Total: 74 T1 (n = 44) T2 (n = 30)	99%	90%	100%	T1 (n = 44) Sensitivity: 100% T2 (n = 30) Sensitivity: 88%
Nimura, H. 2004 [36]	84	99%	100%	67%	ICG and infrared ray electronic endoscopy (IREE)
Park, D.J. 2006 [25]	100	94%	78.6%	100%	FNR occurred in cases with tumor size > 4 cm
Koyama, T. 2007 [37]	50	95%	81%	50%	Low specificity
Kusano, M. 2008 [27]	22	90.9%	T1: 66.7% Overall: 40%	T1: 100% Overall: 100%	T1 stage FNR:33.3%% Overall FNR:60% (T1,T2,T3)
Kelder, W. 2010 [28]	212	99.5%	85.8%	100%	ICG and infrared ray electronic endoscopy (IREE)
Tajima, Y. 2010 [8]	77	LAG: 94.7% OG: 94.9%	LAG: 75% OG: 76.9%	LAG: 100% OG: 100%	LAG: FNR:25% OG: FNR:23.1%
Park, D.J. 2011 [29]	68	91.2%	72.2%	100%	Dual technique (ICG and isotope)
Miyashiro, I. 2013 [30]	241	99.6%	89.7%	98.6%	Intraoperative FNR: 10.3%
Miyashiro, I. 2014 [38]	440	97.8%	85.7%	99.3%	High FNR
Quaresma, L. 2015 [39]	20	92.85%	83.3%	100%	size of the tumor is less than 4 cm
Tummers, Q.R. 2016 [40]	22	95%	75%	100%	FNR occurred in T3 and T4 cases
Takahashi, N. 2017 [41]	44	100%	100%	100%	cT1 gastric adenocarcinomas less than 4 cm
Kim, D.W. 2019 [31]	28	55%	98.9%	76%	Dual tracer method as gold standard
Mayanagi, S. 2020 [42]	132	100%	95.7%	100%	multicenter retrospective cohort study

**Table 2 cancers-14-05143-t002:** Summary of studies compare ICG-FI guided lymph node harvest with non-ICG method in GC.

Ref.	Patients	ICG Group	Non-ICG Group	ICG-Mean (SD) Nodules	Non-ICG-Mean (SD) Nodules	*p* Value
Ushimaru, Y. 2019 [59]	168	84	84	47.5 ± 1.7	42.6±1.7	0.046 *
Kwon, I.G. 2019 [56]	80	40	40	48.9 (14.6)	35.2 (11.2)	<0.001 ***
Chen, Q.Y. 2020 [57]	258	129	129	50.5 (15.9)	42 (10.3)	<0.01 **
Cianchi, F. 2020 [51]	74	37	37	50.8 (17.1)	40.1 (23)	0.03 *
Liu, M. 2020 [52]	136	61	75	33.72 (9.06)	29.36 (8.76)	0.005 **
Ma, S. 2020 [55]	65	31	34	49.6 (13.3)	36.4 (13.3)	0 ***
Huang, Z.N. 2021 [53]	313	102	211	41.5 (12.9)	28.4 (12.4)	<0.001 ***
Lu, X. 2021 [54]	56	28	28	27.50 (10.60)	21.79 (6.73)	0.0196 *
Romanzi, A. 2021 [65]	20	10	10	40	24	<0.05 *
Zhong, Q. 2021 [66]	514	385	129	49.9	42.0	<0.001 ***

Significance (* *p* <  0.05; ** *p*  < 0 .01; *** *p*  < 0 .001).

**Table 3 cancers-14-05143-t003:** Multi-modalities as dual-tracer-guided techniques integrates with ICG-FI.

Integrate Modalities	Rationale	Superiority	Drawbacks
Traditional biological Blue dye			
Patent blue (PB)	Biological dye, visible in white light	Relies solely on a visual identification of the SLNs and no requirement of additional equipment	Restricted dying time and fades quickly when exposed to light
Methylene blue (MB)			Toxicity
Isosulfide blue (ISB)			Allergic reactions and long-term skin discoloration
Traditional radiocolloid tracer			
Technetium-99 (^99m^Tc) [80]	Identification of decays emitting X-rays by gamma probe	Auxiliary SPECT–CT enhance SLN mapping in terms of overall and bilateral detection rates and identify SLNs located in unusual anatomic sites	The dose is time-dependent, the decay of TC-99m may compromise the effectiveness and Radioactive drug (^99m^Tc) toxicity
Fluorescent agent			
5-aminolevulinic acid (5-ALA) [82]	Tumor-specific accumulation of photoactive PpIX	Accumulates specifically in cancer cells and provide fluorescence signal	Diffuse-type tumors shown only nominal red fluorescent signal
Sodium fluorescein (SF) [83]	Excited by 400–550 nm wavelengths light and provide fluorescence signal	The excitation and emission wavelengths of SF do not overlap with ICG, SF can be a remedial measures for contaminated surgical field caused by spillage or incorrect injection of ICG	Potential leak from lymphatic vessels
Carbon nano-tracers			
Carbon nanoparticles [86]	Carbon nanoparticles pass through the lymphatic vessels and accumulate in the lymph nodes, thus staining them black	With average diameter of 150 nm, Carbon nanoparticles only enter lymphatic vessels but not blood capillaries	Carbon nanoparticles remain in the body for a long time, thus extravasation will affect the surgical field
magnetic tracer			
Sienna+^®^ [84]	Magnetic resonance imaging (MRI)	High resolution, ideal tissue penetration depth	Potential adverse including patient discomfort, artifacts on magnetic resonance imaging (MRI) and brown skin staining
Superparamagnetic iron-oxide Nanoparticles (SPIONs) [85]		The integration of ICG avoids higher echelon node involvement during SLN procedure	Skin staining issues; potential unnecessary nodal retrieval if there is no co-localization of sentinel nodes by the two techniques
other			
PET/CT [87]	-	Detect nodes near the primary tumor	High cost and Radioactive drug toxicity

**Table 4 cancers-14-05143-t004:** Intraoperative biopsy technologies that improve diagnostic yield of ICG-FI.

Biopsy Techniques	Rationale	Superiority	Drawback
One-step nucleic acid (OSNA) [98]	mRNA assay that can detect micro metastases in lymph nodes based on cytokeratin 19 (CK19) levels	Highly concordant with standard histology	requires fresh nodal tissue
Transcription-polymerase chain reaction (RT-PCR) [99]	RT-PCR assay, detecting metastases in lymph nodes based on cytokeratin 19 (CK19) or carcinoembryonic antigen (CEA) levels	Detecting micro metastases of lymph nodes	Time cost, requires fresh nodal tissue
Confocal laser endomicroscopy (CLE) [91]	Recognize Tumor-specific fluorescent antibodies at a cellular or tissue level	Provide tumor-specific real-time in vivo immunohistochemistry	Fluorescence signal dependence, unable to detect tissue without fluorescent antibody
Optical coherence tomography (OCT) [92]	Optical imaging at a cellular or tissue level	Real-time, in vivo, no-invasive	The scanning of OCT is insufficient to scan the tissue in depth

## Data Availability

The data presented in this study are available in this article and Appendix A.

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
