# Peer review of "Mapping Lymph Node during Indocyanine Green Fluorescence-Imaging Guided Gastric Oncologic Surgery: Current Applications and Future Directions"

_cancers, 2022, doi:10.3390/cancers14205143_

Round 1
Reviewer 1 Report
Lymphadenectomy is a key moment in the treatment of malignant stomach tumors. The study is interesting. The evaluation of the use of indocyanine green to regulate lymphadenectomy makes the study interesting.Well structured manuscript.
Exhaustive.
Adequate number of references.
Complete and clear tables.
Author Response
Q: Lymphadenectomy is a key moment in the treatment of malignant stomach tumors. The study is interesting. The evaluation of the use of indocyanine green to regulate lymphadenectomy makes the study interesting.
Well structured manuscript.
A: Thank you for your decision and constructive comments on my manuscript.

Reviewer 2 Report
I appreciate your etreme efforts for this article. I think it is concisely described about ICG mapping and navigation surgery for GCs.
The authors have written ICG imaging and navigation surgery in various situations and future directions. Multi-tracer methods are used in breast cancer surgery; therefore, it seems to be perfect.
Not original, but very concisely written, I think this article will earn many citations. That's why I judged as accpet.
The authors have cited previous studies and made discussion in detail.
Suggestions: Larger pictures of figure 2 are better for readers.Tables seems to be very informative.
Author Response
Q: I appreciate your etreme efforts for this article. I think it is concisely described about ICG mapping and navigation surgery for GCs.
The authors have written ICG imaging and navigation surgery in various situations and future directions. Multi-tracer methods are used in breast cancer surgery; therefore, it seems to be perfect.
Not original, but very concisely written, I think this article will earn many citations. That's why I judged as accpet.
The authors have cited previous studies and made discussion in detail.
Suggestions: Larger pictures of figure 2 are better for readers.
A: We appreciate it very much for your good suggestion, and we have done it according to your ideas.

Reviewer 3 Report
The review article, "Mapping lymph node during indocyanine green fluorescence-imaging guided gastric oncologic surgery: current applications and future directions" written by Tang et. al., explains the current approaches and application of ICG-FI and evaluates the potential approaches to enhance the ICG-FI. Although it is a well written review, authors can make few minor changes to improve overall quality of this article. Here are the suggestions.
1. The limitation of ICG-FI could be explained briefly in a new section
2. Concluding remarks could be reframed and explained with few more details.
Author Response
Q: The review article, "Mapping lymph node during indocyanine green fluorescence-imaging guided gastric oncologic surgery: current applications and future directions" written by Tang et. al., explains the current approaches and application of ICG-FI and evaluates the potential approaches to enhance the ICG-FI. Although it is a well written review, authors can make few minor changes to improve overall quality of this article. Here are the suggestions.
- The limitation of ICG-FI could be explained briefly in a new section
- Concluding remarks could be reframed and explained with few more details.
A: Thank you for your rigorous comments, we have revised the manuscript as follow:
- we have explained the limitations of ICG-FI in a new section as your suggested (please kindly find it in revised manuscript Page 5, Line162).
- we have restructured and expanded our concluding remarks in this review.
Again, we appreciate it very much for your good suggestion.
